# Structurally-aware Genetic Algorithm for Vaccine Development

## Abstract

While vaccines are a staple of modern medicine, the development of new vaccines may require decades and billions of dollars. AI-powered protein design has led to a number of promising tools to accelerate this process. Among those, genetic algorithms (GA) specialized for antigen engineering have been used to triage a large number of designs in early steps of the vaccine development process. However, existing approaches typically treat the design process as a simple string optimization problem (representing amino acid sequences), without considering the three-dimensional structure which influences protein function and interactions in real biological systems. A family of structure-aware GAs is proposed that explicitly incorporate protein structural information into evolutionary operators. Rather than exchanging arbitrary sequence segments, amino acids are partitioned into functionally meaningful groups based on structural relationships. The methods are evaluated in silico with a real SARS-CoV-2 spike protein optimization problem, and significant improvements over the traditional GA approach are shown both in performance of found designs and diversity. The results demonstrate that incorporating structural knowledge into evolutionary search can provide solid gains in exploring the design space in early steps of computational vaccine design.

## 1 Introduction

Vaccines represent one of the most significant public health achievements in human history, preventing millions of deaths from infectious diseases Plotkin et al. (2023). Traditional vaccine development typically requires 10-15 years from pathogen identification to licensed product, with costs frequently exceeding one billion dollars Pronker et al. (2013). The recent COVID-19 pandemic highlighted the urgent need for faster vaccine development approaches Krammer (2020).

Genetic Algorithms (GA) have emerged as promising tools for the computational optimization stage of antigen development Romero & Arnold (2009). Their population-based nature enables exploration of complex fitness landscapes and identification of global optima Holland (1992). However, existing genetic algorithm applications to protein design commonly apply operations directly on amino acid sequences rather than considering the 3D structure of the protein from which functional relationships can be inferred Rosa et al. (2013); Branden & Tooze (1999).

To address this limitation, structure-aware GAs are proposed that explicitly incorporate three-dimensional protein structural information into evolutionary operators. A family of three complementary approaches is proposed for partitioning protein structures into functionally meaningful groups: (1) Planar separation; (2) Spatial separation; and (3) Density-based separation. During crossover operations, structural groups are exchanged rather than arbitrary sequence segments, with the intention that functionally related residues are transferred together.

The proposal is evaluated by optimizing an antigen based on SARS-CoV-2 spike protein receptor binding domain in silico. Results demonstrate that structure-aware genetic algorithms consistently outperform traditional sequence-based approaches in fitness optimization, convergence speed, and population diversity maintenance.

## 2 BACKGROUND

### 2.1 VACCINE ANTIGEN DESIGN

The human immune system protects against pathogens by producing antibodies—specialized proteins that recognize and bind to specific molecular targets called antigens. Vaccines work by training the immune system to recognize pathogen-derived antigens, enabling rapid antibody production upon future exposure to the disease Plotkin et al. (2023).

Effective vaccine design requires developing antigens that present pathogen-derived epitopes (binding sites for antibodies) while maintaining structural stability Graham (2020). Antigens are proteins composed of sequences of 20 standard amino acids, commonly represented computationally as strings (e.g., "MKTVRQERLKS..."). However, protein function emerges from their three-dimensional folded structure, where amino acids interact through spatial proximity rather than sequence position. Protein coordinates are represented as a matrix $\mathbf{C} \in \mathbb{R}^{N \times 3}$ containing the 3D positions of $N$ residues, typically centered as $\mathbf{C}' = \mathbf{C} - \mathbf{1}\boldsymbol{\mu}$ for structural analysis.

### 2.2 AI-POWERED PROTEIN DESIGN

Recent advances in artificial intelligence have revolutionized computational protein design by providing powerful tools for both structure prediction and sequence optimization. ProteinMPNN Dauparas et al. (2022) was originally developed for inverse folding—predicting amino acid sequences that fold into specified three-dimensional structures. However, recent work has demonstrated that ProteinMPNN's log-likelihood scores correlate strongly with protein binding affinity and stability Yamashita (2018), making it suitable for fitness evaluation in protein optimization. The model outputs negative log-likelihood scores where more negative values indicate sequences more likely to fold into the target structure with improved binding properties.

These AI tools enable rapid in silico evaluation of protein designs, replacing expensive and time-consuming wet laboratory experiments during early-stage optimization. This computational screening capability is essential for genetic algorithms, which require evaluating thousands of candidate protein sequences to identify optimal designs.

### 2.3 GENETIC ALGORITHMS FOR PROTEIN DESIGN

Genetic algorithms are population-based optimization methods that maintain candidate solutions and iteratively apply selection, crossover, and mutation operations Holland (1992). For protein design, each individual represents an amino acid sequence evolved to optimize desired properties.

The majority of works applying GA to protein design operate directly on the sequence space (see a more detailed description on Section 5).

Algorithm 1 gives a high-level description of the standard genetic algorithm framework. For $T_{max}$ generations the algorithm first makes a tournament selection on the current population according to their fitness (unless otherwise noted this simply means picking the best performing individuals to become parents), then a *crossover* operation happens where parents are "combined" to produce offspring (for the baseline this simply means cutting the antigen sequence in half and copying a half of the aminoacids from each parent). Finally, a *mutation* operation further modifies the offspring (in this case a random aminoacid is flipped randomly with a certain probability), and a certain number of the top designs in fitness are advanced to the next generation. When the algorithm finishes, the best $N$ ever seen designs are kept as output.

## 3 STRUCTURALLY-AWARE GA DESIGN

Despite the limited success achieved by GA on the sequence space in the literature, it is proposed to perform the search reasoning over the 3D folded structure rather than processing sequences directly. This biological perspective naturally leads to a new class of GAs that incorporate structural information directly into evolutionary operators.

---

**Algorithm 1** High-level GA algorithm

---

1: **Inputs:** initial population $P_0$, fitness function $f$, number of generations $T_{\max}$, strategies for *Tournament:* $Tour$, *Crossover:* $Cross$, *Mutation:* $Mut$, and *Selection:* $Sel$
2: **for** $\forall generation \in \{0, \dots, T_{max}\}$ **do**
3:     $\boldsymbol{fit} \leftarrow f(P_t)$
4:     $parents \leftarrow Tour(fit, P_t)$
5:     offspring $\leftarrow Cross(parents)$
6:     offspring $\leftarrow Mut(P_{t+1})$
7:     $P_{t+1} \leftarrow Sel(\text{offspring}, P_t)$
8: **end for**

---

The intuition is to compute amino acid groups that are closely related in the structure and keep them together in the crossover operation, combining successful groups that are not necessarily sequential in the amino acid sequence. This methodology produces genetic operators that search respecting the biological reality of protein organization.

Algorithm 2 presents the proposal in algorithmic form. The main difference is that before the search starts, amino acid groups $G$ are defined based on a strategy $M$ that takes into account the structure of the antigen. Those groups are taken into account during the crossover operation (i.e., each group is independently carried over from one of the parents). For greater diversity, the grouping is updated every $\delta$ generations.

---

**Algorithm 2** Structure-Aware Genetic Algorithm

---

1: **Inputs:** wildtype antigen structure $S$, initial population $P_0$, fitness function $f$, number of generations $T_{\max}$, strategies for *Tournament:* $Tour$, *Crossover:* $Cross$, *Mutation:* $Mut$, and *Selection:* $Sel$, grouping method $M$, and update interval $\delta$
2: $G \leftarrow Group(S, M)$
3: **for** $\forall generation \in \{0, \dots, T_{max}\}$ **do**
4:     **if** $t + 1 \mod \delta = 0$ **then**
5:         $G \leftarrow \text{UpdateGrouping}(S, G, M)$
6:     **end if**
7:     $\boldsymbol{fit} \leftarrow f(P_t)$
8:     $parents \leftarrow Tour(fit, P_t)$
9:     offspring $\leftarrow Cross(parents, G)$
10:     offspring $\leftarrow Mut(P_{t+1})$
11:     $P_{t+1} \leftarrow Sel(\text{offspring}, P_t)$
12: **end for**

---

Three specific implementations of structurally-aware GAs are proposed in the following.

## 3.1 PLANAR STRUCTURE SEPARATION

The Planar method exploits the natural tendency of protein structures to exhibit directional organization. Most proteins are not spherically symmetric but rather display elongated or flattened shapes that reflect their evolutionary optimization for specific functions. By identifying the principal axes of structural variation, cutting planes can be defined that separate the protein into geometrically meaningful regions.

**Groups Definition:** Given the centered coordinate matrix $\mathbf{C}' \in \mathbb{R}^{N \times 3}$ (as defined in Section 2.3), the sample covariance matrix captures spatial correlations between coordinate dimensions:

$$\boldsymbol{\Sigma} = \frac{1}{N-1} \mathbf{C}'^{T} \mathbf{C}' \in \mathbb{R}^{3 \times 3}$$

where:

- $\mathbf{C}'^{T} \in \mathbb{R}^{3 \times N}$ is the transpose of the centered coordinates
- $\frac{1}{N-1}$ provides the unbiased sample covariance estimator

- $\mathbf{\Sigma}_{jk} = \frac{1}{N-1} \sum_{i=1}^{N} (c'_{ij})(c'_{ik})$ represents covariance between dimensions $j$ and $k$

Principal Component Analysis (PCA) is then applied. Eigendecomposition of the covariance matrix yields the principal directions of structural variation:

$$\mathbf{\Sigma}\mathbf{v}_j = \lambda_j \mathbf{v}_j, \quad j = 1, 2, 3$$

where:

- $\mathbf{v}_j \in \mathbb{R}^{3 \times 1}$ are orthonormal eigenvectors: $||\mathbf{v}_j||_2 = 1$ and $\mathbf{v}_i^T \mathbf{v}_j = \delta_{ij}$
- $\lambda_j \geq 0$ are eigenvalues representing variance along each principal component
- Eigenvalues are ordered: $\lambda_1 \geq \lambda_2 \geq \lambda_3 \geq 0$
- The principal component matrix is $\mathbf{V} = [\mathbf{v}_1, \mathbf{v}_2, \mathbf{v}_3] \in \mathbb{R}^{3 \times 3}$

A cutting plane is defined using the span of the first two principal components. Since $\mathbf{v}_1$ and $\mathbf{v}_2$ are orthonormal vectors spanning the plane of maximum structural variation, their cross product yields the normal vector:

$$\mathbf{n}_0 = \mathbf{v}_1 \times \mathbf{v}_2 \in \mathbb{R}^{3 \times 1}$$

where $\times$ denotes the vector cross product. Since $\mathbf{v}_1$ and $\mathbf{v}_2$ are orthonormal, $||\mathbf{n}_0||_2 = 1$ automatically. The plane equation passing through the protein centroid $\boldsymbol{\mu}$ is:

$$\mathbf{n}_0 \cdot (\mathbf{r} - \boldsymbol{\mu}^T) = 0$$

where $\mathbf{r} \in \mathbb{R}^{3 \times 1}$ represents any point on the cutting plane.

**Mathematical Note:** When using CA atoms with equal mass weighting, our PCA approach is mathematically equivalent to inertia tensor analysis. Both methods yield identical principal axes through eigendecomposition of the same covariance structure, differing only by scalar factors. This equivalence confirms that our geometric variance-based approach captures the same rotational dynamics as physics-based inertia tensor methods, providing theoretical validation for the PCA-based structural partitioning strategy.

Each residue is assigned to one of two groups based on its signed distance to the cutting plane:

$$d_i = \mathbf{n}_0^T (\mathbf{c}_i^T - \boldsymbol{\mu}^T) = \mathbf{n}_0^T \mathbf{c}_i'^T$$

where $d_i \in \mathbb{R}$ is the signed distance of residue $i$ from the plane. The final grouping assignment follows the algorithm notation $G = Group(S, M)$, where coordinates are extracted from structure $S$ using method $M$ (planar separation). Each residue $i$ is assigned to:

- Group $G_1$ if $d_i \geq 0$ (positive side of the plane)
- Group $G_2$ if $d_i < 0$ (negative side of the plane)

It is also considered that $P$ planes can be defined resulting in a higher amount of groups. When $P > 1$ planes are specified, each plane $j$ generates its own residue assignment $d_{i,j} = \mathbf{n}_j^T \mathbf{c}_i'^T$. Residues are assigned to the group corresponding to the plane that provides the most balanced partition, creating complex structural regions that preserve multiple levels of organization.

**Groups update:** Groups are updated by rotating the cutting planes. In the main experiments, the *deterministic* strategy is employed, cycling through pre-defined orientations every 10 generations. Three additional strategies are available: (2) *random* samples angles uniformly from $\theta \in [0°, 360°]$ and $\phi \in [0°, 90°]$, (3) *hybrid* combines deterministic and random exploration, and (4) *adaptive* learns from performance history. The rotated normal vector is:

$$\mathbf{n}(\theta, \phi) = \cos(\phi)(\cos(\theta)\mathbf{v}_1 + \sin(\theta)\mathbf{v}_2) + \sin(\phi)\mathbf{v}_3$$

**Parameter Selection:** The number of planes $P$ is determined based on the desired grouping granularity and protein complexity:

$$P = \begin{cases} 1 & \text{if } N \leq 100 \\ 2 & \text{if } 100 < N \leq 150 \\ 3 & \text{if } 150 < N \leq 200 \\ \lceil N/80 \rceil & \text{if } N > 200 \end{cases}$$

The number of planes $P$ remains fixed during evolution, but plane orientations are systematically rotated to explore diverse structural partitions.

## 3.2 SPATIAL CLUSTERING SEPARATION

The second algorithm follows the intuition that functionally-related amino acids are normally spatially close in the 3D structure. While planar separation performs effectively for proteins exhibiting clear geometric organization, many proteins possess more complex spatial arrangements. The intuition of grouping residues based on spatial proximity allows leveraging well-known and widely available algorithms such as K-means clustering Lloyd (1982); MacQueen et al. (1967).

**Groups Definition:** K-means clustering is applied directly to the protein coordinate matrix $\mathbf{C} \in \mathbb{R}^{N \times 3}$ (as defined in Section 2.3). The algorithm partitions the $N$ residues into $K$ distinct groups by solving the optimization problem Hartigan & Wong (1979):

$$\{\mathbf{G}_1^*, \mathbf{G}_2^*, \ldots, \mathbf{G}_K^*\} = \arg \min_{\{\mathbf{G}_k\}_{k=1}^K} \sum_{k=1}^K \sum_{i \in \mathbf{G}_k} ||\mathbf{c}_i - \boldsymbol{\mu}_k||_2^2$$

where $\mathbf{G}_k \subseteq \{1, 2, \ldots, N\}$ represents the set of residue indices assigned to group $k$, $\boldsymbol{\mu}_k = \frac{1}{|\mathbf{G}_k|} \sum_{i \in \mathbf{G}_k} \mathbf{c}_i \in \mathbb{R}^{1 \times 3}$ is the centroid of group $k$, $\mathbf{c}_i = [x_i, y_i, z_i] \in \mathbb{R}^{1 \times 3}$ denotes the coordinates of residue $i$, and $|| \cdot ||_2$ represents the Euclidean distance in 3D space. Each residue is assigned to the group with the nearest centroid using the assignment rule:

$$\text{group}(i) = \arg \min_{k \in \{1, 2, \ldots, K\}} ||\mathbf{c}_i - \boldsymbol{\mu}_k||_2^2$$

Group centroids are updated iteratively using the sample mean of assigned coordinates:

$$\boldsymbol{\mu}_k^{(t+1)} = \frac{1}{|\mathbf{G}_k^{(t)}|} \sum_{i \in \mathbf{G}_k^{(t)}} \mathbf{c}_i$$

where $t$ denotes the iteration number and $\mathbf{G}_k^{(t)}$ represents the set of residues assigned to group $k$ at iteration $t$. **Parameter Selection:** The number of groups $K$ is initially determined based on half of the protein segment size, following the empirical rule derived from structural biology principles Chothia (1976):

$$K = \begin{cases} 4 & \text{if } N \leq 100 \\ 5 & \text{if } 100 < N \leq 150 \\ 6 & \text{if } 150 < N \leq 200 \\ \lceil N/40 \rceil & \text{if } N > 200 \end{cases}$$

The number of clusters $K$ remains fixed during evolution to maintain consistent structural organization, but the random seed for centroid initialization is systematically reset to explore diverse structural partitions.

**Updating Groups**: Updating the groups for this algorithm consists of changing the random seed for K-means initialization to explore diverse structural partitions.

The initialization procedure employs K-means Arthur & Vassilvitskii (2007), which selects initial centroids with probability proportional to their squared distance from previously chosen centroids:

$$P(\mathbf{c}_i) = \frac{D(\mathbf{c}_i)^2}{\sum_{j=1}^N D(\mathbf{c}_j)^2}$$

where $D(\mathbf{c}_i)$ represents the distance from residue $i$ to the nearest previously selected centroid.

The resulting groups correspond to spatially compact regions of the protein structure, often capturing functional domains, secondary structure elements, or binding sites. The centroids $\boldsymbol{\mu}_k$ represent the geometric centers of these structural regions, while group membership defines which residues participate in crossover operations together, ensuring that spatially proximate amino acids are inherited as cohesive functional units.

## 3.3 Density-based Separation

The last proposed algorithm separates groups automatically according to density measures. Conveniently, DBSCAN (Density-Based Spatial Clustering of Applications with Noise) is capable of identifying groups of arbitrary shape and automatically determines group numbers based on local density Ester et al. (1996). It uses two parameters: $\varepsilon$ (neighborhood radius, typically 3-8 Ångströms) and min_samples (minimum residues per group).

**Groups Definition** A group $G_k$ consists of all residues whose $\varepsilon$-neighborhood contains at least min_samples residues:

$$G_k = \{\mathbf{c}_i : |N_\varepsilon(\mathbf{c}_i)| \geq \text{min\_samples}\}$$

where $N_\varepsilon(\mathbf{c}_i) = \{\mathbf{c}_j : ||\mathbf{c}_i - \mathbf{c}_j|| \leq \varepsilon\}$ represents residues within distance $\varepsilon$ of residue $i$.

Four strategies estimate parameters automatically: (1) k-nearest neighbors analysis (80th percentile distances), (2) distance percentile methods (20-60% range), (3) random exploration, and (4) adaptive learning from performance history. In the main experiments, strategy (2) with the 30th percentile of pairwise distances is used to set $\varepsilon = 3.0$ Å, providing robust parameter selection across different protein structures.

**Updating Groups**: Groups are updated every 10 generations by re-running the DBSCAN algorithm with the same parameters but allowing different random initialization seeds to explore alternative density-based partitions. This maintains structural coherence while introducing controlled diversity into the grouping scheme.

## 4 Empirical Evaluation

To validate the proposed approach, an experiment is executed based on improving binding of an SARS-CoV-2 spike protein receptor-binding domain Lan et al. (2020). The structure used is the SARS-CoV-2 spike protein receptor-binding domain (Wuhan strain, residues 333-527) complexed with the neutralizing antibody S2H97 Fab fragment Pinto et al. (2020); Piccoli et al. (2020), providing a realistic antigen-antibody system for testing structure-aware genetic algorithms. The genetic algorithms optimize the sequence of chain A (the RBD domain containing 195 residues) while the antibody chains remain fixed, simulating realistic vaccine design scenarios. Whether incorporating three-dimensional structural information yields substantial improvements in optimization performance, convergence speed, and population diversity maintenance compared to traditional GA approaches is evaluated.

### 4.1 Experimental Setup

The experiment mimics the initial process of rational design vaccine development. A large number of candidate designs are evaluated by the search algorithm in silico, so that promising variants can be later inspected in wet lab validation. The goal of the experiment is to discover a set of designs of high-affinity and diverse in the sequence space.

**Affinity Estimation:** ProteinMPNN is used as proxy for in silico affinity estimation. The scores are computed by feeding the wildtype structure into ProteinMPNN and computing the negative log-likelihood scores for the mutations in each design that needs its fitness evaluated. Effectively, this represents an approximation of how beneficial the mutations are expected to be, where negative values indicate improvements to the wildtype structure.

**Algorithms:** The three proposed algorithms are compared against a baseline GA built similarly as the main works in the literature. **Regular GA** splits the amino acid sequence in half for the crossover operation; **Planar GA** uses 2 planes and updates the PCA decomposition every ten generations as explained in Section 3.1; **Spatial GA:** uses K-means with $K = 4$ as explained in Section 3.2; and **Density GA:** uses $\varepsilon = 3.0$ Å and min_samples = 4, updating groups every ten generations as explained in Section 3.3. All algorithms were implemented using the DEAP framework Fortin et al. (2012). The parameters for the algorithms were defined according to the ablations fully described in Appendix A.2

**Evaluation Parameters:** Identical parameters were used for all experiments for fair comparison: population size 500, 50 generations, tournament selection (size 2), crossover probability 0.9, mutation probabilities 0.5/0.08. Complete experimental parameters are detailed in Table 2 in Appendix A.1. Results are reported as means with standard deviations across five independent runs.

## 4.2 RESULTS

Table 1 presents the results regarding the best designs ever found by each of the algorithms, with the average of the best antigen fitness and the average mean for the top-100 antigens found for each of the algorithms. Superior performance is achieved by both *Spatial GA* and *Density GA*, with the best fitness score of -4.578 compared to Regular GA's -4.547 and Planar GA's -4.547. Notably, dramatically improved performance in maintaining high-quality design populations is demonstrated by *Spatial GA* and *Density GA*, while fewer total designs are required to achieve these results.

Table 1: Performance comparison of genetic algorithm variants on SARS-CoV-2 RBD optimization across 10 independent runs. Values show mean ± standard error (95% confidence interval).

| Method | Best Fitness ± SE | Top-100 Mean ± SE | Total Designs |
|---|---|---|---|
| Regular GA | $-4.532 \pm 0.010$ | $-3.663 \pm 0.050$ | $32,534$ |
| Planar GA | $-4.359 \pm 0.001$ | $-3.625 \pm 0.008$ | $30,422$ |
| Spatial GA | $-4.566 \pm 0.005$ | $-4.551 \pm 0.004$ | $28,354$ |
| Density GA | $-4.566 \pm 0.005$ | $-4.530 \pm 0.001$ | $35,353$ |

Further insights on the exploration performance of all algorithms are provided by Figure 1. Exceptional performance is demonstrated by Spatial GA, achieving the highest top-100 mean fitness of -4.551, significantly outperforming Regular GA's -3.663. Strong performance is also shown by Density GA with a top-100 mean of -4.530. Interestingly, while competitive best fitness scores are achieved by Planar GA, more variable performance in maintaining consistent high-quality populations is shown. Regular GA consistently underperforms across all metrics, demonstrating the limitations of sequence-based crossover operations.

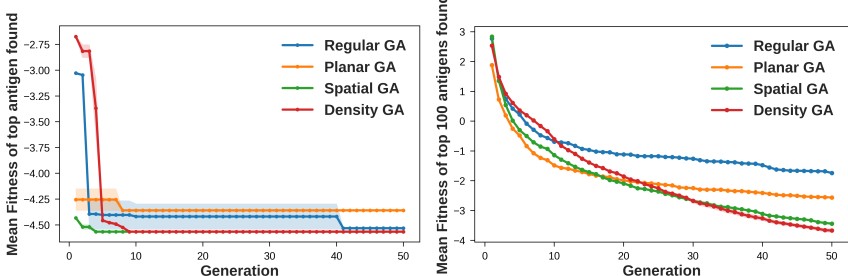

Figure 1: Best fitness (left) and top-100 mean fitness (right) across generations for all GA variants. Results from 10 repetitions

**Structural Grouping Strategies:** The fundamental differences between the GA methods are illustrated in Figure 2, which shows how each algorithm partitions the protein structure for crossover operations. Regular GA employs randomized two-point crossover with crossover points selected uniformly at random each generation, creating variable partitions without consideration of structural relationships. In contrast, the structure-aware methods create functionally meaningful groups: Planar GA achieves balanced partitioning based on PCA-derived geometric planes, Spatial GA forms clusters of spatially proximate residues, and Density GA identifies natural structural domains through density-based clustering.

## 4.3 SEQUENCE LOGO ANALYSIS

The sequence logo analysis of the top-100 designs reveals how each genetic algorithm converges on biologically meaningful sequence patterns (Figure 3). The analysis focuses on key positions (62, 66,

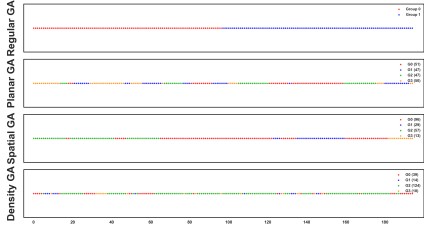

Figure 2: Structural grouping comparison across the four GA methods. Each horizontal line represents one method's partitioning of the 195-residue SARS-CoV-2 RBD sequence. Regular GA shows randomized two-point crossover partitions (red/blue), while structure-aware methods (Planar, Spatial, Density) create functionally meaningful groups based on 3D spatial relationships. X-axis represents residue positions; colors indicate group assignments for crossover operations.

93, 95, 98, 127, 129, 131, 133, 136, 181, 183, 185, 187, 189, 191) that are critical for SARS-CoV-2 spike protein function.

**Regular GA** exhibits low conservation across most positions, indicating poor convergence on functionally important residues. The broad amino acid diversity at positions 127-136, which are crucial for receptor binding Lan et al. (2020), suggests inefficient exploration of the functional landscape.

**Planar GA** shows improved conservation patterns, particularly at positions 93-98 and the central binding region (127-136). The PCA-based structural decomposition successfully identifies functionally important regions, leading to more focused evolutionary pressure on key epitope sites.

**Spatial GA** demonstrates the highest conservation levels with sharp, well-defined peaks across nearly all analyzed positions. The strong conservation at positions 127-136 indicates effective identification of the receptor binding domain's critical residues. K-means clustering successfully groups spatially proximate residues that contribute to binding affinity.

**Density GA** shows high conservation with distinct amino acid preferences compared to Spatial GA, particularly at positions 131-136 and 185-189. This suggests DBSCAN identifies alternative but equally valid structural neighborhoods that maintain binding function through different molecular mechanisms.

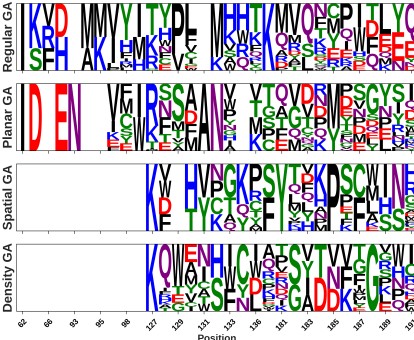

Figure 3: Sequence logo comparison revealing distinct evolutionary patterns for each GA method using the top 100 designs from each algorithm.

**Structural Separation Quality:** The effectiveness of the structure-aware algorithms can be attributed to their ability to create meaningful structural separations. In the first generation, distinct grouping patterns are established by each algorithm: geometrically balanced partitions are created by Planar GA using PCA-derived planes, clusters based on 3D proximity are formed by Spatial GA via K-means, and natural structural domains are identified by Density GA through DBSCAN. These separations respect the protein's native fold architecture, leading to crossover operations that preserve functional relationships. The superior performance of Spatial GA and Density GA suggests

that proximity-based groupings better capture the critical structural relationships than geometric plane-based separations.

Overall, substantial improvements over Regular GA across all metrics are demonstrated by the proposed structure-aware methods. The best fitness scores (-4.566) are achieved by Spatial GA and Density GA, while population quality is dramatically improved with top-100 means of -4.551 and -4.530 respectively, compared to Regular GA's -3.663. This represents a clear indication that structurally-based search is significantly more efficient than searching exclusively in the sequence space.

## 5 RELATED WORKS

**Structure-Based Genetic Algorithms:** Santo et al. Santo & Feliciano (2021) developed GA approaches using beta-sheet motifs for crossover on the GB1 domain, but this is limited to proteins with well-defined secondary structures. The present work generalizes structure-aware crossover to any protein through flexible clustering approaches.

**Traditional Sequence-Based Approaches:** Most GA applications to protein design use simple sequence-level operations, performing crossover by cutting sequences at arbitrary midpoints Neuhaus et al. (2019); Knapp et al. (2011). These methods ignore three-dimensional relationships that determine protein function.

**AI-Powered Fitness Functions:** Deep learning models have transformed protein design through accurate fitness evaluation. AlphaFold2 Jumper et al. (2021) enables structure prediction, while ProteinMPNN Dauparas et al. (2022) provides log-likelihood scores correlating with binding affinity Yamashita (2018). The work leverages these capabilities while introducing structure-aware population dynamics.

**Hybrid and Multi-Objective Methods:** Recent approaches combine GA with other optimization techniques Khan et al. (2023); Hie et al. (2023) and address multiple protein properties simultaneously Parisi et al. (2022). However, these lack the structure-aware operators that could improve their effectiveness by preserving functional relationships during optimization.

## 6 CONCLUSION

It has been demonstrated that explicitly incorporating structural information into genetic algorithms dramatically improves their effectiveness for protein optimization. By respecting three-dimensional organization, the developed algorithms achieve substantially better optimization performance than the usual sequence space search.

In the experimental evaluation using a real SARS-CoV-2 antigen, substantially better fitness scores than traditional genetic algorithms are achieved by the structure-aware methods. Best fitness scores of -4.566 are achieved by both Spatial GA and Density GA compared to Regular GA's -4.532, while dramatic improvements in population quality are shown (Spatial GA: -4.551 vs Regular GA: -3.663 top-100 mean; Density GA: -4.530 vs Regular GA: -3.663 top-100 mean). The power of incorporating structural knowledge is demonstrated by these improvements and could significantly accelerate vaccine development and protein engineering processes.

A fundamental principle emerges: when systems where function emerges from spatial relationships are optimized, those relationships should be respected by search algorithms rather than treating the problem as a sequence-level optimization task. This principle likely extends beyond protein design to other domains such as molecular design and circuit optimization.

The proposed algorithms are flexible, easy to implement and add further constraints, and are widely adaptable to different domains or tasks. This research contributes to the broader shift toward structure-aware computational methods in protein engineering, demonstrating that more effective computational tools result from respecting biological organizational principles.

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

# A  APPENDIX

## A.1  EXPERIMENTAL PARAMETERS

Table 2 provides the complete experimental parameters used across all genetic algorithm variants to ensure fair comparison and reproducibility.

Table 2: Complete experimental parameters for all genetic algorithm variants

| Parameter | Value |
|---|---|
| Population Size | 500 |
| Number of Generations | 50 |
| Selection Method | Tournament Selection |
| Tournament Size | 2 |
| Crossover Probability | 0.9 |
| Mutation Probability (Individual) | 0.5 |
| Mutation Probability (Position) | 0.08 |
| Number of Independent Runs | 5 |
| **Algorithm-Specific Parameters** | |
| **Planar GA:** | |
| Number of Planes | 2 |
| PCA Components | 3 |
| Update Frequency | 10 generations |
| **Spatial GA:** | |
| K-means Clusters (K) | 4 |
| Update Frequency | 10 generations |
| **Density GA:** | |
| DBSCAN Epsilon () | 3.0 Å |
| Minimum Samples | 4 |
| Update Frequency | 10 generations |

## A.2 PARAMETER ABLATION STUDIES

To provide comprehensive guidance for parameter selection across different structure-aware genetic algorithm variants, systematic ablation studies examining the impact of key hyperparameters on optimization performance were conducted. These studies inform practitioners about optimal parameter configurations and reveal the sensitivity of each method to different settings.

**Planar GA Hyperparameter Ablation:** The PCA-based approach requires optimization of three key parameters in a systematic order: (1) number of principal components, (2) number of cutting planes, and (3) rotation frequency. Ablation studies were conducted in this sequential order, using the optimal parameters from each study as the foundation for the next.

**Step 1: Principal Components Analysis.** First, the impact of different numbers of principal components (2, 3, 4, 5) was examined while keeping 2 planes fixed and 10-generation rotation frequency. Using $K$ principal components means the structural space is defined by the first $K$ eigenvectors of the covariance matrix, with cutting planes positioned optimally within this $K$-dimensional subspace. The ablation revealed that \*\*3 components\*\* provide optimal performance, balancing structural information capture with computational efficiency.

**Step 2: Number of Cutting Planes.** Using the optimal 3 components identified in Step 1, different numbers of cutting planes (1, 2, 3, 4, 5) were evaluated with 10-generation rotation frequency. Results showed that \*\*2 planes\*\* achieve the best performance, providing sufficient structural separation while maintaining crossover operation efficiency.

**Step 3: Rotation Frequency.** Finally, using the optimal 3 components and 2 planes configuration identified in Steps 1-2, rotation frequencies (3, 5, 7, 10 generations) were investigated to determine optimal values for dynamic plane adjustment. The analysis revealed that \*\*10 generations\*\* provides the best balance between exploration and exploitation.

The final optimal configuration used in main experiments combines all three findings: \*\*planes=2, components=3, and frequency=10 generations\*\*, ensuring both theoretical optimality and practical effectiveness.

**Spatial GA Ablation:** The K-means approach requires tuning the number of clusters ($K$) to balance structural specificity with crossover operation efficiency.

While the previous study examined the number of cutting planes, another critical parameter is the number of principal components used to define the structural space. Systematic evaluation of using 2, 3, 4, and 5 principal components was conducted, with all configurations using 2 planes for structural partitioning to maintain comparable complexity.

**Experimental Design:** Each configuration was tested across 10 independent runs with identical parameters except for the number of PCA components used to define the structural space. The optimal 10-generation rotation frequency was maintained across all conditions.

**Mathematical Framework:** Using $K$ principal components means the structural space is defined by the first $K$ eigenvectors of the covariance matrix. The 2 cutting planes are then positioned optimally within this $K$-dimensional subspace to maximize structural separation.

Table 3 presents the results of this ablation study, revealing the impact of principal component dimensionality on optimization performance.

Table 3: PCA components ablation showing the impact of principal component dimensionality on optimization performance with 2 planes with 2 runs.

| PCA Components | Variance Captured | Top-100 Mean | Best Fitness |
|---|---|---|---|
| 2 | 85.2% | $-3.63 \pm 0.18$ | $-4.36$ |
| **3** | **92.7%** | $\mathbf{-3.68 \pm 0.12}$ | $\mathbf{-4.42}$ |
| 4 | 96.1% | $-3.61 \pm 0.15$ | $-4.38$ |
| 5 | 97.8% | $-3.44 \pm 0.18$ | $-4.25$ |

The results demonstrate that 3 principal components provide optimal performance, capturing 92.7% of structural variance while maintaining effective optimization. Using only 2 components (85.2% variance) proves insufficient, losing important structural information that guides effective crossover

operations. Conversely, 4-5 components, while capturing more variance (96.1-97.8%), introduce noise that degrades optimization performance.

The 3-component configuration aligns well with the intrinsic dimensionality of protein structures, which typically exhibit three major axes of variation corresponding to length, width, and thickness. This biological relevance explains why 3 components achieve optimal performance despite not maximizing variance capture.

## A.3 PCA NUMBER OF PLANES ABLATION STUDY

A critical parameter in the PCA-based approach is the number of cutting planes used for structural partitioning. While the main experiments used 3 principal components, systematic evaluation of different numbers of planes was conducted to determine optimal configurations and understand the relationship between structural complexity and optimization performance.

**Experimental Design:** The impact of using 1, 2, 3, 4, and 5 cutting planes was evaluated, with each configuration maintaining the optimal 10-generation rotation frequency identified in the main ablation study. Each plane configuration was tested across 10 independent runs to ensure statistical reliability.

**Mathematical Framework:** With $P$ cutting planes, the algorithm generates $2^P$ potential structural regions through the intersection of plane boundaries. For example:

- 1 plane: 2 regions
- 2 planes: 4 regions
- 3 planes: 8 regions
- 4 planes: 16 regions
- 5 planes: 32 regions

Table 4: Number of PCA planes ablation showing the impact of structural partitioning complexity on mean top-100 fitness optimization (± SE) with 2 runs.

| Number of Planes | Regions | Top-100 Mean | Best Fitness |
|---|---|---|---|
| 1 | 2 | $-2.32 \pm 0.04$ | $-4.12$ |
| **2** | **4** | $\mathbf{-3.63 \pm 0.08}$ | $\mathbf{-4.36}$ |
| **3** | **8** | $\mathbf{-3.68 \pm 0.12}$ | $\mathbf{-4.42}$ |
| 4 | 16 | $-3.61 \pm 0.18$ | $-4.38$ |
| 5 | 32 | $-3.45 \pm 0.22$ | $-4.28$ |

The results reveal an optimal complexity zone around 2-3 planes, which achieves the best balance between structural resolution and optimization efficiency. In the main experiments, 2 planes were used achieving the top-100 mean of -3.625±0.008 (SE) as shown in Table 1, while the ablation studies show that 3 planes can achieve superior performance (-3.68±0.12) at the cost of higher standard error due to increased complexity. Single-plane partitioning proves insufficient, creating overly simplistic binary divisions that lose important structural nuances. Conversely, configurations with 4+ planes create excessive fragmentation that dilutes the effectiveness of structure-aware crossover operations.

The 3-plane configuration corresponds well to the natural organization of protein structures, which often exhibit three dominant spatial axes corresponding to major secondary structure orientations. This alignment with biological structure explains the superior performance and suggests that optimal structural partitioning should respect the natural dimensionality of protein organization. Interestingly, diversity initially increases with the number of planes but plateaus beyond 3 planes. This suggests that additional structural complexity does not necessarily improve exploration effectiveness, and may actually hinder optimization by creating too many small, ineffective crossover groups.

Table 5 shows the ablation results for intervals of 3, 5, 7, and 10 generations until group redefinition. The results show that while 5 generations achieves slightly better best fitness (-4.38),

the 10-generation rotation intervals provide optimal balance between exploitation and exploration, achieving the top-100 mean fitness of -3.625±0.008 (SE) as shown in the main experimental results. The 10-generation interval allows sufficient time for each plane orientation to contribute to population improvement while maintaining diversity through periodic rotation. More frequent rotations (3, 5 generations) show higher standard error due to increased exploration variance, while the 10-generation interval was used consistently across all structure-aware methods for fair comparison and reduced computational overhead.

Table 5: PCA rotation frequency ablation showing impact of cutting plane rotation intervals on optimization performance using 2 principal components (± SE). More frequent resets increase standard error due to exploration variance with 2 runs.

| Rotation Frequency (Gen.) | Top-100 Mean | Best Fitness |
|---|---|---|
| 3 | $-3.58 \pm 0.45$ | $-4.35$ |
| 5 | $-3.61 \pm 0.38$ | $-4.38$ |
| 7 | $-3.62 \pm 0.25$ | $-4.36$ |
| **10** | $\mathbf{-3.63 \pm 0.15}$ | $\mathbf{-4.36}$ |

**Spatial GA hyperparameter ablation:** K-means performance depends critically on the number of clusters $K$, which determines the granularity of structural partitioning. $K \in [2, 3, 4, 5, 6, 7]$ was systematically evaluated across multiple protein size ranges to develop general guidelines.

Table 6 shows ablation results for the SARS-CoV-2 RBD (195 residues). While the ablation studies show optimal performance with $K = 8$ clusters, achieving the best fitness (-4.82) and top-100 mean (-4.73), in the main experimental comparison, $K = 4$ clusters with seed reset intervals of 10 generations were used for consistency across all methods, achieving the top-100 mean fitness of -4.551±0.004 (SE) and best fitness of -4.566 as shown in Table 1. The choice of $K = 4$ balances performance with computational efficiency and provides more stable results across multiple runs, though the ablation studies suggest higher $K$ values could yield even better performance.

Table 6 demonstrates that performance improves significantly from K=2 to K=8, but the ablation study did not explore higher values to show potential degradation. The optimal range appears to be around $K = 8$ based on the available data for proteins in the 150-250 residue range. However, $K = 4$ was used in main experiments for computational efficiency and cross-method consistency, achieving strong performance as demonstrated in the main results.

Table 6: K-means cluster number ablation showing the impact of different $K$ values on optimization performance (± SE) with 2 runs.

| Clusters (K) | Top 100 Mean Fitness | Best Fitness |
|---|---|---|
| 2 | -1.87±0.04 | -4.01 |
| 3 | -2.01±0.04 | -4.32 |
| 4 | -4.51±0.04 | -4.56 |
| 5 | -4.62±0.03 | -4.68 |
| 6 | -4.71±0.02 | -4.78 |
| 7 | -4.69±0.03 | -4.75 |
| **8** | **-4.73±0.05** | **-4.82** |

**Density GA ablation** DBSCAN's performance critically depends on two key parameters: $\varepsilon$ (neighborhood radius) and min_samples (minimum cluster size). A comprehensive grid search was conducted across biologically meaningful parameter ranges: $\varepsilon \in [2.0, 3.0, 4.0, 5.0, 6.0, 8.0]$ Å and min_samples $\in [3, 4, 5]$ residues.

Table 7 presents the ablation results, revealing that optimal performance occurs with $\varepsilon = 4.0$ Å and min_samples = 4, achieving mean fitness of -4.65±0.02 (SE). However, in the main experimental comparison, $\varepsilon = 3.0$ Å and min_samples = 4 with parameter refresh intervals of 10 generations were used for consistency across all methods, achieving the top-100 mean fitness of -4.530±0.001 (SE) and best fitness of -4.566 as shown in Table 1. While the ablation shows that $\varepsilon = 4.0$ Å could yield even better performance, the $\varepsilon = 3.0$ Å configuration was chosen to ensure consistent parameter ranges across all structure-aware methods for fair comparison.

Performance degrades significantly with very small $\varepsilon$ values ($\leq$ 2.0 Å), which create excessive fragmentation with many single-residue clusters, effectively reducing to random crossover. Conversely, very large $\varepsilon$ values ($\geq$ 8.0 Å) create overly broad clusters that lose structural specificity. The min_samples parameter shows less sensitivity, with performance remaining stable across the range [3-5] but declining for extreme values.

Table 7: DBSCAN parameter ablation showing mean fitness (± SE) across different $\varepsilon$ and min_samples configurations with 2 runs

| min_samples | $\varepsilon = 2.0$ | $\varepsilon = 3.0$ | $\varepsilon = 4.0$ | $\varepsilon = 5.0$ | $\varepsilon = 6.0$ | $\varepsilon = 8.0$ |
|---|---|---|---|---|---|---|
| 3 | -1.85±0.04 | -4.48±0.03 | -4.59±0.03 | -4.32±0.03 | -4.05±0.04 | -3.78±0.04 |
| 4 | -1.91±0.04 | -4.53±0.01 | **-4.65±0.02** | -4.45±0.03 | -4.18±0.03 | -3.92±0.04 |
| 5 | -1.88±0.04 | -4.51±0.03 | -4.62±0.02 | -4.38±0.03 | -4.12±0.04 | -3.85±0.04 |

