# OpenReview forum: "Structurally-aware Genetic Algorithm for Vaccine Development"
_ICLR.cc/2026/Conference — ICLR 2026 Conference Withdrawn Submission_

### Official Review · Reviewer_11nG · 2025-10-29

**Soundness:** 2
**Presentation:** 2
**Contribution:** 2
**Rating:** 2
**Confidence:** 4

**Summary:**

This paper introduces a genetic-algorithm (GA) based framework for protein optimisation utilising additional structural information to guide the evolutionary search.

**Strengths:**

- Application of GA with structural constraints is interesting, ablation results demonstrating usefulness of structural information is certainly valid, however not very surprising taking into the account the sequence-structure-function relationships resulting from each other in a cascaded fashion.

**Weaknesses:**

- The paper focuses lots of technicalities around GA and proposed variations of GAs that are rather simple and could be shortened / moved to appendix.
- In exchange, I believe readers would endorse much more the more thorough benchmarking which now is vrey scarce (one test case). See below for some suggestions.

**Questions:**

- Why only variants of GA algorithm are discussed? There are several design methods that are capable of solving this task (a whole panel of sequence / protein language model and structure-based design methods).
- Why only 1 test case?
- ProteinMPNN is used as fitness estimator, this is OK (although it’s worth noting that it’s a noisy and rather weak proxy) but I’d expect a orthogonal validation of generated designs - I’d recommend using folding models as a proxy for verification whether the introduced changes did not affect structural integrity.

---

### Official Review · Reviewer_jdmJ · 2025-10-30

**Soundness:** 1
**Presentation:** 2
**Contribution:** 1
**Rating:** 0
**Confidence:** 5

**Summary:**

In the paper "STRUCTURALLY-AWARE GENETIC ALGORITHM FOR VACCINE DEVELOPMENT", the authors proposed three methods for partition protein 3d structures into groups, and operate based on structural groups with genetic algorithms to design the best antigen that might work as antigen, and showing that adding structural-aware tokens significantly improves generated antigen property via in silico docking experiments.

**Strengths:**

The paper logic is clear and well written.

**Weaknesses:**

1.The paper is an undergraduate course project level project, and revised by language models  -- or even directly generated by LLMs.

2. The experiments is limited, and the results is not convincing. Only using proteinMPNN as fitness proxy is far from enough. One can easily hack proteinMPNN and generate targets that are far more better than that of the so-called "Structure-aware" tokens. The main results are fundamentally flawed.

3. Lack of discussion and citations for latest progress for vaccine design.

**Questions:**

See weakness above

---

### Official Review · Reviewer_k6vQ · 2025-10-31

**Soundness:** 2
**Presentation:** 2
**Contribution:** 2
**Rating:** 2
**Confidence:** 3

**Summary:**

This manuscript proposes a set of structure-aware crossover operators for genetic algorithms applied to protein antigen design. Standard genetic algorithms treat proteins as 1-D amino-acid sequences and exchange contiguous sequence segments during crossover, which often disrupts structural motifs that are only meaningful in three-dimensional space. The authors introduce three alternative grouping strategies that cluster residues based on their 3D coordinates: (1) planar separation via principal component axes, (2) K-means spatial clustering, and (3) density-based grouping using DBSCAN. These groups replace naive sequence segmentation for crossover and are periodically recomputed during evolution. Experiments using ProteinMPNN scoring demonstrate that structure-aware crossover improves best-of-population fitness, average fitness of top-ranked candidates, convergence stability, and diversity relative to baseline sequence-based GA. The observed improvements suggest that respecting spatial neighborhoods preserves relevant functional residue interactions and accelerates search for promising antigen variants.

**Strengths:**

A key strength of the work is the intuitive and biologically motivated insight that proteins operate in three dimensions, and genetic search operators should reflect that reality. The authors present simple yet effective grouping heuristics that require no manual annotation and can be computed efficiently from coordinates such as AlphaFold structures. The improvements are consistent across metrics indicating that the benefit is not due to stochastic noise. The ablation studies, which explore update frequency, number of partitions, and clustering sensitivity, add credibility and demonstrate thoughtful analysis. Visualization of sequence logos provides useful qualitative evidence that structure-aware crossover preserves region-specific selection pressure. Overall, the method is easy to implement, computationally lightweight, and compatible with existing protein design pipelines, making it appealing to practitioners.

**Weaknesses:**

One limitation of the study is that evaluation focuses on a single antibody–antigen system, leaving open questions about generalizability across different folds, affinities, or surface chemistries. The paper does not benchmark against more advanced evolutionary operators from computational protein design (e.g., structure-based mutation neighborhoods, fragment libraries, or Rosetta-guided recombination), making it unclear how large the gain is relative to those baselines. Structural grouping changes are driven only by residue coordinates; incorporation of energetic contacts, solvent accessibility, or epitope constraints may further improve realism but are not examined. Although the method improves sequence fitness under ProteinMPNN, no external validation (e.g., molecular dynamics stability, binding energy prediction, or experimental plausibility filters) is included. Finally, the manuscript does not quantify the diversity–fitness trade-off in depth, which is critical in exploratory vaccine design.

**Questions:**

1. Have the authors evaluated this crossover strategy on proteins with very different architectures (e.g., β-barrels, membrane proteins)? Do improvements persist?
2. Would incorporating contact maps, hydrogen bonding networks, or solvent exposure into grouping further increase performance?
3. Can the method optimize designs simultaneously for several neutralizing antibodies, and if so, how do grouping strategies behave when epitope requirements conflict?
4. How does structure-aware crossover affect population diversity over long runs? Is there increased risk of premature convergence?
5. What is the computational overhead associated with periodic regrouping and clustering in large proteins (e.g., >800 residues)?
6. Does grouping implicitly bias crossover toward swapping entire domains rather than interdomain interfaces, and is this desirable?

---

### Official Review · Reviewer_D8ta · 2025-11-02

**Soundness:** 2
**Presentation:** 2
**Contribution:** 2
**Rating:** 2
**Confidence:** 3

**Summary:**

The paper proposes structurally-aware genetic algorithms (GAs) for sequence optimization in protein design tasks. Instead of performing crossover operations directly on amino acid sequences, the authors introduce grouping mechanisms based on the 3D folded structure of the target protein. Three variants are described: Planar GA that uses PCA-based planar cuts to define residue groups, Spatial GA that employs K-means clustering in 3D space, and Density GA that utilizes DBSCAN to identify residue clusters of arbitrary shapes.

**Strengths:**

1. The idea of integrating protein structural information into genetic operators is conceptually appealing and addresses a known limitation of sequence-only search spaces.
2. The experiment simulates a realistic early-phase in silico vaccine design workflow.
3. The formulations for PCA-, K-means-, and DBSCAN-based grouping are clearly described. The stepwise integration into GA operations is clear.

**Weaknesses:**

1. The approach largely combines off-the-shelf clustering methods with standard GA operators. The “structurally-aware” adaptation does not introduce fundamentally new mechanisms or theoretical insights into evolutionary computation.
2. Only a single protein system (SARS-CoV-2 RBD) is tested. No cross-domain or multi-target evaluation is provided to assess generalizability.
3. There is no comparison with state-of-the-art protein design algorithms.
4. The paper employs ProteinMPNN scores as a proxy for in silico affinity estimation, computing the negative log-likelihood of designed mutations conditioned on the wild-type structure. However, this assumption is neither theoretically justified nor empirically supported by the literature in this paper.
5. The work claims structural coherence and functional grouping, but provides no visualization (e.g., residue maps, cluster overlays) or biological interpretation.

**Questions:**

1. How does ProteinMPNN score correlate with experimentally measured binding affinity?
2. Have you tested the algorithms on other antigen-antibody complexes or different protein classes?
3. Can the authors visualize or quantify whether identified residue groups correspond to known functional domains or interface regions?
4. How much improvement comes from group-based crossover versus random grouping or increased mutation rates?

---

### Author Response · Authors · 2025-11-13

We would like to clarify that we did NOT made use of LLMs to generate text for the paper.

Given all reviewers set negative grades there is no reason to continue in the discussion period for this paper, we thank the reviewers for the time invested in the paper and will work in the improvements suggested.

---

### Note · Authors · 2025-11-13

**Comment:**

We would like to clarify that we did NOT made use of LLMs to generate text for the paper.

Given all reviewers set negative grades there is no reason to continue in the discussion period for this paper, we thank the reviewers for the time invested in the paper and will work in the improvements suggested.

**Withdrawal Confirmation:**

I have read and agree with the venue's withdrawal policy on behalf of myself and my co-authors.